# Bone Density May Be a Promising Predictor for Blood Loss during Total Hip Arthroplasty

**DOI:** 10.3390/jcm11143951

**Published:** 2022-07-07

**Authors:** Wei Zhu, Zhanqi Wei, Tianjun Zhou, Chang Han, Zehui Lv, Han Wang, Bin Feng, Xisheng Weng

**Affiliations:** 1Department of Orthopaedics, Peking Union Medical College Hospital, Chinese Academy of Medical Sciences & Peking Union Medical College, Beijing 100730, China; zhuwei9508@163.com (W.Z.); wei-zq15@mails.tsinghua.edu.cn (Z.W.); ztj16@mails.tsinghua.edu.cn (T.Z.); hanchanglc@gmail.com (C.H.); pumc_425155628@student.pumc.edu.cn (Z.L.); hannah0928@hotmail.com (H.W.); 2School of Medicine, Tsinghua University, Haidian District, Beijing 100084, China; 3State Key Laboratory of Complex Severe and Rare Diseases, Peking Union Medical College Hospital, Chinese Academy of Medical Science & Peking Union Medical College, Beijing 100730, China

**Keywords:** bone density, total hip arthroplasty, predictor, blood loss

## Abstract

Background: Total hip arthroplasty (THA), which is performed mostly in elderly individuals, can result in substantial blood loss and thereby imposes a significant physical burden and risk of blood transfusion. The femoral neck cut and reamed acetabulum are the main sites of intraoperative bleeding. Whether the bone density in that region can be used to predict the amount of blood loss in THA is unknown. Methods: We retrospectively analyzed adult patients undergoing primary THA in the Department of Orthopedics, Peking Union Medical College Hospital, from January 2018 to January 2020. All these patients underwent primary unilateral THA. Patients had their bone mineral density (BMD) recorded within the week before surgery and were stratified and analyzed for perioperative blood loss. Multivariable regressions were utilized to adjust for differences in demographics and comorbidities among groups. Results: A total of 176 patients were included in the study. Intraoperative blood loss was 280.1 ± 119.56 mL. Pearson correlation analysis showed a significant correlation between blood loss and preoperative bone density of both the femoral greater trochanter (R = 0.245, *p* = 0.001) and the Ward’s triangle (R = 0.181, *p* = 0.016). Stepwise multiple linear regression showed that preoperative bone density of the greater trochanter (*p* = 0.015, 95% CI: 0.004–0.049) and sex (*p* = 0.002) were independent risk factors for THA bleeding. The area under the receiver operating characteristic curve (AUROC) of the greater trochanter and Ward’s triangle was 0.593 (95% CI: 0.507–0.678, *p* = 0.035) and 0.603 (95% CI: 0.519–0.688, *p* = 0.018), respectively. The cutoff T value on the femoral greater trochanter for predicting higher bleeding was −1.75. Conclusions: In THA patients, preoperative bone density values of the femoral greater trochanter and sex could be promising independent predictors for bleeding during surgery. Osteoporosis and female patients might have lower blood loss in the THA operation.

## 1. Introduction

Total hip arthroplasty (THA) is currently considered to be the most effective treatment for end-stage hip arthritis and is increasingly being used worldwide. During THA, there can be substantial perioperative blood loss; an average drop of 3.5–4 g/dL in hemoglobin has been reported [1]. In previous studies, the perioperative transfusion rate in primary THA was between 15% and 60% [2,3,4]. In a study involving 573 THA patients, the average blood loss was 378.6 mL [5]. Although blood safety has progressed significantly in past years, serious complications, such as nosocomial infections, transfusion-associated lung injury, and circulatory overload have not been eliminated [6,7], which has directed effort toward exploring alternative methods of blood management in the perioperative period. However, powerful indicators that can accurately predict the amount of bleeding during THA are still lacking at present.

In view of the considerable amount of bleeding and inevitable probability of blood transfusion during THA surgery, the discovery of indicators that can predict perioperative blood loss before surgery would be of considerable importance in formulating treatment guidelines. The extent of perioperative bleeding could then be estimated in advance, which would be helpful for perioperative intervention or blood transfusion. In a study involving 524 patients undergoing hip surgery for developmental dysplasia of the hip, lower preoperative hemoglobin, lower body mass index (BMI), and bilateral surgery were associated with an increased risk of perioperative blood transfusion [8]. However, this study focused on hip surgery in children, and there is a lack of research on hip surgery in adults for predicting bleeding situations. Komnos et al. [9] reported that the anterior approach could reduce the bleeding during THA compared with the posterolateral approach. However, the posterolateral approach is still the mainstream THA method because of advantages such as its shorter operation time, clearer anatomical structure, and shorter learning curve [10].

In a prospective randomized clinical study, Pesce et al. [11] analyzed the results of bone mineral density (BMD) evaluation in hip surgery and underlined the importance of BMD evaluation by the region of interest in order to underline the importance of this parameter both post- and preoperatively. A high proportion of patients undergoing THA are elderly. However, bone quality is gradually lost with age. The thinning of and reduction in the bone trabeculae and the stratification and thinning of the bone cortex are some of the manifestations of osteoporosis [12]. Whether these changes in bone quality are related to intraoperative blood loss remains to be studied.

During the THA procedure, the bone at the level of the femoral neck cut and at the level of the reamed acetabulum ooze blood. Therefore, we speculated that bone density in that region would affect the amount of bleeding during hip replacement. The results could help clinicians predict the amount of perioperative blood loss and take intervention measures to intervene.

## 2. Patients and Methods

### 2.1. Patients

We retrospectively studied patients diagnosed with end-stage hip disease, such as femoral head necrosis (FHN) or developmental dysplasia of the hip (DDH), who underwent THA by a single senior surgeon between January 2018 and September 2020. All patients were enrolled in our study and signed an informed consent form before inclusion. This study was approved by the institutional review board (IRB) of Peking Union Medical College Hospital (NO. S-K1425), and the flow is shown in Figure 1.

The inclusion criteria included adult patients with a diagnosis of femoral head necrosis, primary hip arthritis, secondary hip arthritis, or developmental dysplasia of the hip, who required THA. The exclusion criteria included patients requiring revision hip arthroplasty; patients diagnosed with femoral neck or intertrochanteric fracture; patients diagnosed with hip joint infection; patients for whom bone density examination was not performed; patients diagnosed with other diseases such as coagulation disorders, immune system diseases, serious heart diseases, or tumors; and patients on long-term anticoagulation, antiplatelet, or hormone therapy before surgery.

### 2.2. General and Biomedical Characteristics

Clinical data, including age, sex, and BMI, were obtained through a retrospective chart review. BMD was measured on the patients’ femoral neck, Ward’s triangle, greater trochanter, and lumbar spine after the patient was hospitalized and within one week before the operation (Figure 2). Ward’s triangle at the femoral head and neck is the junction area between the medial trabecular bone under pressure and the lateral trabecular bone under tension, forming a triangular vulnerable area. With dual-energy X-ray absorptiometry, the photon peaks pass through the body, and the scanning system sends the signals to a computer for data processing to determine bone mineral content. The result is expressed in terms of the T value.

After admission, routine blood measurements were performed both before and after surgery to determine the erythrocyte-specific volume. These biochemical parameters were routinely measured in the hospital central laboratory by skilled technical personnel. Then, total blood volume (TBV) and intraoperative blood loss (BV) were calculated according to the formulas reported in the literature [13]. The specific formulas are as follows [14]:


*Male:*
Total blood volume (TBV) = 0.3669 × height × height × height + 0.03219 × weight + 0.6041 
Blood loss volume = TBV × (preoperative erythropoietic volume − postoperative erythropoietic volume)


*Female*:TBV = 0.3561 × height × height × height + 0.03308 × weight + 0.1833Blood loss volume = TBV × (preoperative erythropoietic volume − postoperative erythropoietic volume)

### 2.3. Surgical Procedure

All hip surgeries were primary THAs performed by a single senior surgeon. The posterolateral hip joint approach was used for all surgeries, followed by femoral neck cut and then reamed acetabulum. All hip prostheses were performed using a Pinnacle acetabular and Corail uncemented femoral stem by DePuy. During the operation, each patient was intravenously injected with 1 g of tranexamic acid prior to surgical skin incision. Anticoagulant therapy was started on the first day and continued for five weeks. On the first day after the operation, patients stood and performed functional exercise as soon as possible.

### 2.4. Statistical Analysis

Student’s *t*-test was used for intergroup comparisons. For nonnormally distributed data, a nonparametric test was used for intergroup comparisons. Intergroup comparisons of categorical variables were performed using the chi-squared test. Pearson correlation analysis was conducted to evaluate the association between parameters. Stepwise multiple linear regression analysis was performed to investigate factors independently associated with blood loss volume during THA surgery. All tests were two-sided, and *p* < 0.05 was considered statistically significant.

SPSS software version 20.0 (SPSS, Chicago, IL, USA) was used for all statistical analyses.

## 3. Results

### 3.1. General Characteristics

A total of 176 patients were included in the study (Table 1), including 59 (33.5%) males and 117 (66.5%) females. These patients were 54.7 ± 15.77 years old and had a mean intraoperative blood loss of 280.1 ± 119.56 mL. The male and female patients showed no significant difference in age or BMI (*p* > 0.05). As shown in Table 1, male patients had obviously higher bone mineral density in the lumbar nadir (*p* = 0.008), femoral Ward’s triangle (*p* = 0.023), and greater trochanter (*p* = 0.001); however, there was no significant difference in the femoral neck (*p* = 0.084). This revealed that in the middle-aged and elderly patient groups, the BMD of men was significantly higher than that of women. In THA, the intraoperative blood loss in men was 328.1 ± 150.76 mL (*n* = 59), which was significantly higher than that in women (255.9 ± 91.85 mL, *n* = 117, *p* = 0.001).

### 3.2. The Association between Bone Mineral Density and Blood Loss Volume

The results from Pearson correlation analysis are shown in Figure 3. In these patients, there was no significant correlation between blood loss and preoperative BMD for the femoral neck (R = 0.078, *p* = 0.304, Figure 3C), but there was a strong positive correlation between blood loss and preoperative BMD for femoral Ward’s triangle (R = 0.181, *p* = 0.0161, Figure 3E) and total femoral (R = 0.178, *p* = 0.018). BMD in the greater trochanter of the femur was positively correlated with the amount of blood loss during surgery (R = 0.245, *p* = 0.001, Figure 3F). For BMD in other places of the body, regardless of the T value of lumbar nadir (R = 0.089, *p* = 0.241) and total lumbar (R = 0.095, *p* = 0.212), there was no significant relationship with intraoperative blood loss.

### 3.3. Bone Mineral Density Predicts Blood Loss

Stepwise multiple linear regression analysis was performed on 176 patients undergoing unilateral THA, and the results are shown in Table 2. Preoperative femoral greater trochanter BMD (*p* = 0.015) and sex (*p* = 0.002) were each independently associated with intraoperative blood loss. THA patients were divided by mean intraoperative blood loss into a low-blood-loss group and a high-blood-loss group. ROC curves are shown in Figure 4 for the femoral neck (2A) with AUCROC = 0.577 (95% CI: 0.493–0.661, *p* = 0.078), the greater trochanter (2B) with AUCROC = 0.633 (95% CI: 0.552–0.715, *p* = 0.002), and Ward’s triangle (2C) with AUCROC = 0.621 (95% CI: 0.538–0.704, *p* = 0.006). The cutoff T value of the femoral greater trochanter was −1.75.

## 4. Discussion

Although the current use of tranexamic acid has greatly reduced the amount of blood loss and the transfusion rate in THA [15], blood loss remains substantial. The most important finding of this study is that preoperative BMD can predict blood loss volume during THA, which had rarely been reported in previous studies [16,17]. The results of this study show that the average amount of blood loss in 176 unilateral THA patients was 280.1 ± 119.56 mL. Because the patients are mostly elderly and in poor health [18], this relatively high blood loss imposes a significant physical burden. The ability to find predictors for THA bleeding can be used for targeted individual intervention before surgery to ensure the perioperative safety of patients and reduce the probability of blood transfusion [19,20,21].

Using the Pearson correlation coefficient test, we found that the bleeding volume in THA patients had a strong positive significant correlation with bone density in the total femur (R = 0.178, *p* = 0.018, Figure 4C), femoral Ward’s triangle (R = 0.181, *p* = 0.016, Figure 3E), and femoral greater trochanteric area (R = 0.245, *p* = 0.001, Figure 3F). With an increase in the T value, local bone density increased, leading to lower bleeding during surgery.

This study found that not all preoperative BMD values effectively predicted the amount of bleeding in THA. As shown in Figure 3 and Figure 4, only the preoperative BMD of the greater trochanter and total femur could predict blood loss in THA. We speculate that bone density might affect the amount of bleeding at the femoral neck cut and reamed acetabulum. Increased local BMD might indicate rich local intraosseous vascular conditions, leading to increased perioperative blood loss. In Table 2, multiple linear regression analysis showed that sex and femoral greater trochanter bone-mineral density were independent risk factors for predicting the amount of blood loss during surgery. The results showed that the minimum and average values of lumbar bone density could not accurately predict the amount of blood loss during surgery (Figure 3 and Table 2). Therefore, for patients undergoing primary THA, targeted preoperative interventions can be carried out in advance if the patient has a higher preoperative BMD of the femoral greater trochanter.

During a retrospective analysis of 21,239 patients, Sarah et al. [22] found that the preoperative international normalized ratio (INR) value was an independent factor in predicting initial unilateral knee replacement bleeding. Hyun-Jung et al. [23] studied the factors influencing hemorrhage in hip revision surgery, and by analyzing data from 58 cases, they found that maximum clot firmness of FIBTEM could be used as an index to predict perioperative hemorrhage. However, a study [24] of factors influencing bleeding after total hip replacement published in 2003 found that female patients with higher systolic blood pressure and lower bone density had lower blood loss after hip replacement. The sample size of this study was small, and the amount of postoperative blood loss was studied, which was consistent with the surgical bleeding in our study. Another study [25] including 1048 patients found that in joint replacement, people older than 70 were more likely to bleed, which was an independent risk factor, although the authors did not study the relationship between bone density and surgical bleeding. Our study has some suggestive significance for future clinical decision making. Preoperative BMD can predict blood loss volume during THA operation. The lower bone density in the femoral greater trochanter indicated lower bleeding during THA. The cutoff T value on the femoral greater trochanter for predicting higher bleeding was −1.75. Sex was also an independent risk factor for predicting bleeding during THA. Based on the results, men experienced more bleeding than women during the operation.

In conclusion, this clinical cohort study found that greater trochanter osteoporosis and male sex were associated with more intraoperative bleeding during THA. These findings have the potential to help clinicians more accurately predict intraoperative blood loss in patients undergoing THA. Clinicians should be aware that male patients or patients with greater trochanter osteoporosis are high-risk groups and recommend that patients receive more reasonable and effective interventions during the perioperative period, such as the preoperative use of hemoglobin to enhance hemochrome, the preoperative prestorage of autologous blood, intraoperative blood salvage, and postoperative timely blood transfusion.

However, there are still some limitations to our study. First, the sample size was not large enough, leading to deviations in some indicators. For example, the rate of allogeneic transfusion was 1.13% in patients with relatively low BMD; the rate was 1.86% in patients with relatively high BMD. Second, we evaluated perioperative blood loss during THA through factors, such as the erythropoietic volume, weight, and height of patients, which may have led to bias caused by fluid loss and other factors. Third, our research was carried out in a single center, and its results would be more convincing if it could be carried out in multiple centers. Fourth, elderly patients, especially those with osteoporosis, often have reduced bone quality, and changes in bone quality might also affect bleeding in THA. Therefore, further clinical studies are needed to clarify the possible relationship between bone quality and bleeding.

## 5. Conclusions

In THA patients, preoperative bone density values of the femoral greater trochanter and sex could be independent factors to predict bleeding during surgery. Patients with osteoporosis and female patients had lower blood loss during the THA operation.

## Figures and Tables

**Figure 1 jcm-11-03951-f001:**
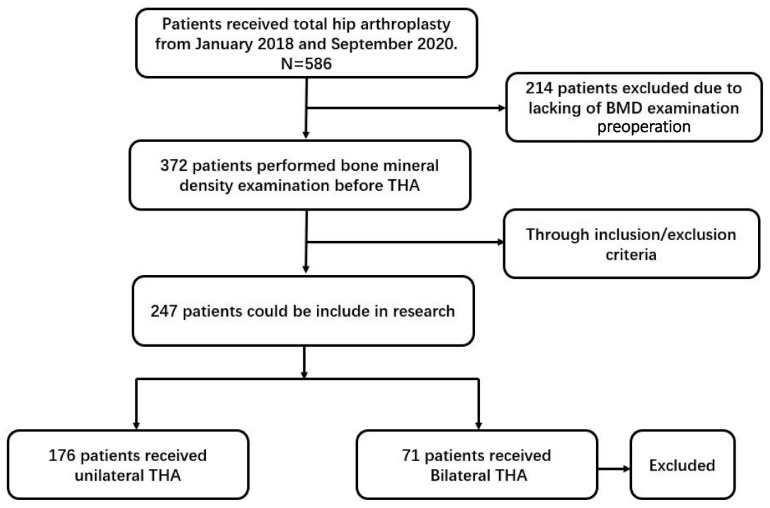
Study flow of this research. BMD is bone mineral density. THA is total hip arthroplasty.

**Figure 2 jcm-11-03951-f002:**
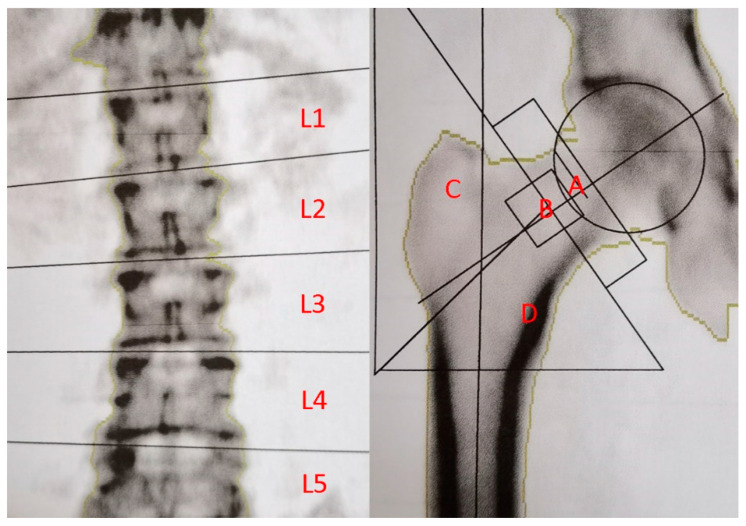
BMD measurement area in the lumbar spine and femur. A: Femoral neck area in BMD. B: Ward’s triangle of the femur area in BMD. C: The greater trochanter of femur area in BMD. D: Femoral shaft area in BMD. BMD is bone mineral density.

**Figure 3 jcm-11-03951-f003:**
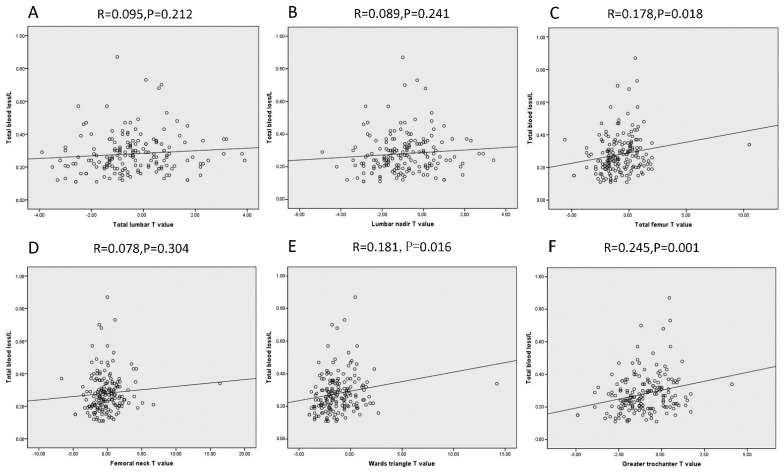
(**A**–**C**): Association between total blood loss volume and total lumbar T value, lumbar nadir T value, and total femoral T value. (**D**–**F**): Association between total blood loss volume and femoral neck T value, Ward’s triangle T value, and femoral greater trochanter T value.

**Figure 4 jcm-11-03951-f004:**
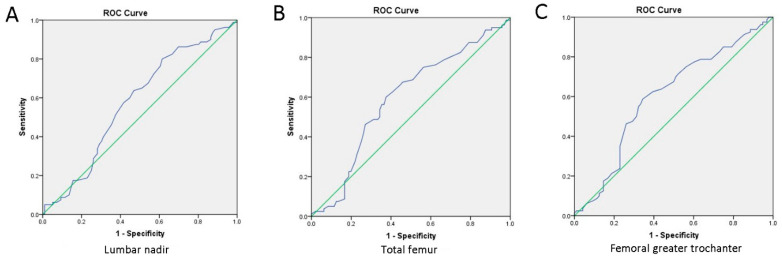
ROC curves for bone density in THA patients. Panel (**A**): AUCROC 0.577 (95% CI: 0.492–0.661, *p* = 0.08). Panel (**B**): AUCROC 0.593 (95% CI: 0.507–0.678, *p* = 0.035). Panel (**C**): AUCROC 0.603 (95% CI: 0.519–0.688, *p* = 0.018).

**Table 1 jcm-11-03951-t001:** Clinical characteristics of the patients undergoing THA.

	Sum (*n* = 176)	Male (*n* = 59)	Female (*n* = 117)	*p* Value
Age (years)	54.7 ± 15.77	53.05 ± 17.28	55.54 ± 14.97	0.325
BMI (kg/m^2^)	24.123 ± 4.168	24.64 ± 4.21	23.86 ± 4.14	0.24
Blood loss volume(mL)	280.1 ± 119.56	328.1 ± 150.76	255.9 ± 91.85	0.001
T value (Total femoral)	−0.779 ± 1.645	−0.289 ± 1.076	−1.026 ± 1.823	0.001
T value (Femoral neck)	−0.158 ± 2.238	0.263 ± 1.651	−0.365 ± 2.463	0.084
T value (Wards triangle)	−1.221 ± 1.855	−0.774 ± 1.432	−1.447 ± 2.004	0.023
T value (Greater trochanter)	−1.043 ± 1.341	−0.564 ± 1.115	−1.285 ± 1.383	0.001
T value (Total lumbar)	−0.429 ± 1.446	−0.124 ± 1.465	−0.567 ± 1.412	0.046
T value (Lumbar nadir)	−0.943 ± 1.382	−0.554 ± 1.415	−1.139 ± 1.329	0.008

BMI: body mass index. The T value was measured using the dual-energy X-ray absorptiometry method and represents the bone mineral content. The *p* value was the comparison between male patients and female patients.

**Table 2 jcm-11-03951-t002:** Stepwise multiple linear regression analysis for the association between preoperative biomedical parameters and surgery blood loss volume in THA patients.

	All Subjects (*n* = 176)			
	β Value	Std β Value	T Value	*p* Value
T value (greater trochanter)	0.026	0.0105	2.47	0.015
T value (femoral neck)	−0.0127	−0.00734	−1.74	0.085
T value (Ward’s triangle)	0.00977	0.0102	0.96	0.339
T value (Total lumbar)	0.032	0.022	1.41	0.161
T value (Lumbar nadir)	−0.039	0.024	−1.61	0.109
BMI (kg/m^2^)	−0.00311	0.00214	−1.45	0.148
Sex	0.0604	0.0189	3.2	0.002
A blood type OR not	−0.0872	0.114	−0.77	0.444
B blood type OR not	−0.0395	0.115	−0.34	0.732
O blood type OR not	−0.048	0.113	−0.42	0.672

BMI, body mass index.

## Data Availability

Not applicable.

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
