# Peer review of "Bone Density May Be a Promising Predictor for Blood Loss during Total Hip Arthroplasty"

_jcm, 2022, doi:10.3390/jcm11143951_

Round 1

Reviewer 1 Report

The paper addresses the very important problem of blood loss during THA. In addition, the authors investigate the clinical problem not only descriptively but also analytically in connection with the assessment of bone quality. The article is interesting and clinically important, but requires a few corrections:

Introduction - the authors should present the clinical problem of bone quality loss already in the introduction. they focus mainly on blood loss, moreover

: Line only in end -stage? - I recommend clarification

Line 54 - it is worth referring to the anterior approach when it comes to possible benefits of its use

134 and 135 add the percentage in the group of women and men

68 add initials

In the discussion, emphasize the possible relationship of bone quality and bleeding, especially in risk groups - one two sentences

Reviewer 2 Report

I feel this manuscript does not offer the reader very much useful information. The amoint of measured blood loss in the low bone density group is not that much different that blood loss mearued in the more dense femoral bone patients. There is no descrption of transfusion requirements and if that differed in the two groups and mor importantly the surgery time was not recorded nor the degree of difficulty such as hip joint deformity, scar formation and other facttors that would add to the possible increase or decrease in blood loss.

Reviewer 3 Report

These authors present a study examining the role of pre-operative bone mineral density in predicting blood loss in patients undergoing total hip arthroplasty. They retrospectively reviewed 176 patients with osteonecrosis of the femoral head, secondary hip arthritis, or developmental dysplasia of the hip, all of whom had preoperative BMD screening one week prior to unilateral THA and recorded blood loss. An unmatched and matched analysis was carried out, and the authors determined that increased BMD of the greater trochanter and male gender were risk factors for increased blood loss, with female gender and osteoporosis were protective against blood loss. This article presents an interesting idea of looking at bone quality as a way to predict blood loss after THA and has the benefit of a large patient series with recent pre-operative BMD screenings.

Despite these strengths there are significant limitations to the manuscript in its current form which I believe limits it’s ability for publication. First, the manuscript needs extensive grammatical and syntax editing – edits by a native English speaker would help the readability of this manuscript significantly. Additionally, there are methodological concerns which I feel limits the utility of this publication. The authors look at BMD as a predictor of intra-operative blood loss, however only in the setting of AVN, secondary arthritis, or DDH of the hip, all of which are different conditions and may have different BMD readings. Thus, it is unclear whether this truly reflects BMD or could reflect that one of these conditions simply predisposes patients to more blood loss. A breakdown of each type of arthritis/condition as well as analysis to control for this would be important. Additionally, there is no control group in this study (such as primary hip osteoarthritis), which should be added as well. The authors do demonstrate that increased greater trochanter osteoarthritis and male gender are associated with increased EBL, however it is also unclear if this is clinically relevant or translates into any changes in outcomes such as need for transfusion. Ultimately, these present significant concerns which limits the ability of this manuscript to be published.

Reviewer 4 Report

The item of the research can be interesting and useful for the readers and for future research on that field.  The statistic section is good.  But the paper is not well written, the english language is not acceptable, several basic mistakes are reported in the manuscript.  The article cannot be accepted in this form.

Some incongruencies are the following:

1) Why is arthroplasty for primary osteoarthritis not included?

2) Why did you not consider the values of haemoglobin and haematocrit? 

3) Line 119: " Anticoagulant therapy was given on the first day after treat-ment."  It sounds like the therapy was given only for one day.

4) Several parts (i.e. lines 137-138, lines 150-151) are repetition of the previous text and should be put in the methods section.

5) Line 201: "the cutting of the greater trochanter was required during hip replacement".  You cut the greater trochanter in hip arthroplasties???

Reviewer 5 Report

Dear Authors the topic is estremely interesting.

As regards the introduction from line 47 to line 59, i suggest to improve the concept linked to role of densitometric evaluation and hip surgery. In fact it is important to underline the importance of BMD evaluation by ROI area in order to underline the importance of this parameter both post and pre operative. For this reason i suggest to cite the following article in which the Authors analyzed the results of BMD evaluation in hip surgery.

The effect of hydroxyapatite coated screw in the lateral fragility fractures of the femur. A prospective randomized clinical study.

Pesce V, Maccagnano G, Vicenti G, Notarnicola A, Moretti L, Tafuri S, Vanni D, Salini V, Moretti B.J Biol Regul Homeost Agents. 2014 Jan-Mar;28(1):125-32.PMID: 24750798 Clinical Trial.”

As regards M&M the strategy  and methods are well described.

As regards the discussion is supported by results.

As regards the conclusion the Authors underlined the importance of BMD evaluation pre operativly in order to stratify the bleeding risk.

Round 2

Reviewer 1 Report

No other comments

Author Response

Your comments make us realize the inadequacy of our work and also bring us very important inspirations. Thank you very much!

Reviewer 2 Report

the manuscript still does not include the operaetive time for the surgeries, if operative time was greater in the normal denisty patients that cfould be the explanation of the increased blood loss! 

>at very least explain why operating time was not included

Line 39 should read 'end stage hip arthritis" not arthroplasty

Reviewer 3 Report

The authors present a revised version of their manuscript examining the role of bone mineral density and its relation to blood loss. I commend the authors on the significant revisions to improve the readability and conclusions made. I would still recommend evaluating the blood loss and need for post-operative transfusion, as this is ultimately why we care about blood loss. Additionally, I would caution the authors on concluding that more aggressive transfusion be pursued especially without information on post-operative transfusion, as we know that blood transfusion increases risk of periprosthetic infection. However, the authors have made significant improvements to the manuscript. Please see minor edits below to aid the readability:

Line 2: would keep “may” instead of changing to “might”

Line 13: would remove “the” before the world “elderly”

Line 15: this idea of femoral and acetabular “broken ends” is brought up throughout the manuscript. “Broken ends” implies a traumatic process not a surgical one, the more correct way to address these would be the “cut femur” or “femoral neck cut” and “reamed acetabulum.” I would change this throughout the manuscript

Line 23: would remove “in the patients”

Line 39: change to “end-stage hip arthritis”

Line 45: change to “nosocomial infections, transfusion-associated lung injury, and circulatory overload have not been eliminated”

Line 47: change to “blood management in the perioperative period”

Lines 59-62: this study presented has no context. Either introduce the concept of hydroxyethyl starch or remove

Line 69: ROI is abbreviated here but not used again in the manuscript so can remove the abbreviation

Line 75: see comment about “broken end”

Lines 78-79: change to “measures to intervene”

Lines 94-95: change “revision of the hip joint or femoral head replacement” to “revision hip arthroplasty”

Line 101: remove the word “collection”

Line 132: see comment about femoral broken end above

Line 136: what does “without incision drainage” mean? Would consider removing

Line 206: remove “operation”

Line 222: see comment about broken end above

Line 257: what drugs specifically to enhance blood levels. Also would refer to as “hemoglobin”

Lines 257-258: some of these seem like expensive and aggressive measures to consider given the fact we don’t actually KNOW that this increases need for transfusion. I would talk about this as a limitation to the study and probably not suggest that these interventions are applied now. For now all I think these authors can say is that surgeons should be aware of this high risk group and counsel patients regarding this perioperatively.

Lines 259-266: Also limitation is not knowing if this clinically matters. Do these patients need more transfusions? Also, only one approach was used which makes it less generalizable.

Reviewer 4 Report

The paper has been re-submitted with some necessary corrections.  Other points have to be improved.

Line 75: During the process of THA. Change into: During the THA procedure

Lines 75-76: the broken end around the greater trochanter and acetabulum... Change into: the bone at the level of the osteotomy of the femoral neck and at the level of the reamed acetabulum

Methods: Why don't you include primary osteoartrhitis?

Lines 94-95: revision of the hip joint or femoral head replacement. Change into: revision surgery

General and biomechanical characteristics: Why, other than the erythrocyte volume, haven't you assessed haemogloblin and haematocrit values?

Line 133: braekage. Use osteotomy.

Line 134: from dePuy. Better "by DePuy."

Lines 175-176: BMD in the greater trochanter of the femur was most strongly correlated with the 175 amount of blood loss during surgery

It should be better described.  It is not well understandable.  Perhaps, we want to specify that, even for the the greater trochanter BMD, the correlation of blood loss is positive. Specify please that, higher the BMD, higher the blood loss! 

Lines 216-217: With a decrease 216 in the T value, local bone density increased,  

Not clear.  Why, if the T-value decreases, should the bone density increase???

Lines 223-224: Increased local BMD might indicate poor local intraosseous vascular conditions, leading to increased perioperative blood loss.

You should better explain this concept: a reduced vascularity, should reduce automatically the bleeding! Is it the opposite?
